# Driving the pulse of the economy or the dilution effect: Inflation impacting economic growth

Piumi Atigala[1], Tharaka Maduwanthi[1], Vishmi Gunathilake[1], Sanduni Sathsarani[1], Ruwan Jayathilaka[2]*

1 SLIIT Business School, Sri Lanka Institute of Information Technology, Malabe, Sri Lanka, 2 Head - Department of Information Management, SLIIT Business School, Sri Lanka Institute of Information Technology, Malabe, Sri Lanka

* ruwan.j@sliit.lk

**Data Availability Statement:** All Data files are available from Central Bank of Sri Lanka (https://www.cbsl.gov.lk) and Department of Census and Statistics, Sri Lanka (http://www.statistics.gov.lk).

## Abstract

Economic growth becomes a critical component in the development of every country since it enhances living standards and other related concerns while eliminating poverty. As a developing country, Sri Lanka must place more emphasis to achieve sustainable economic growth. In addition, various factors have positive and negative impacts on economy's growth. As such, the specific goals of any economy are to sustain long-term economic growth and low inflation. As a result, generally, high inflation is destructive for an economy and low inflation is beneficial. Therefore, it is worth investigating the impact of inflation on economic growth concerning a stable inflation level. This study examines the impact of inflation on economic growth in Sri Lanka by employing the Auto Regressive Distributed Lag model as the estimation technique. Furthermore, the findings illustrate a negative relationship between inflation and economic growth in the short run; when inflation increases by 1%, economic growth decreases by United States Dollar (USD) 3,427.94 million and long run economic growth declines by 107,263.8 million USD. Subsequently, with the current economic reality of Sri Lanka, the macroeconomic policies should be adaptable to maintain the stability of the inflation rate for a sustainable economy.

## Introduction

Sri Lanka still remains as a developing nation although being self-sufficient in human and natural resources. However, the country's prolonged low per capita income and high inflation are warning signs that Sri Lanka is not economically prosperous yet. According to Mensah, Allotey [1], the goal of basic macroeconomics is to achieve sustainable economic growth by maintaining a low inflation. In addition, exchange rate, government expenditure, money supply, oil prices and long-term interest rates are the main criteria for inflation in Sri Lanka. Additionally, creating a conducive environment for economic growth, encouraging investors, increasing employment opportunities and creating a better standard of living are all positive factors that contribute to a low inflationary environment. As Bhattarai [2] indicated, maintaining price

**Funding:** The authors received no specific funding for this work.

**Competing interests:** The authors have declared that no competing interests exist.

stability is critical for the long-term growth and development of the economy. Furthermore, as rising inflation adversely affects the economy, identifying the factors that influence inflation is critical for the economy's proactive decision-making.

According to Gregorio [3], inflation has a negative impact on the economic growth of American countries. Similarly, Smyth [4] indicated that when inflation rises to 10%, economic growth decelerates to 0.025% in Germany. Based on these common findings from past literature indicating both inverse and adverse impacts, the present study investigates the impact of inflation on economic growth in Sri Lanka using 21 years of quarterly time series data during 2000–2021. Additionally, this paper contributes to identifying Sri Lanka's unstable inflation scenario, enabling economic policymakers to make the right economic decisions.

## Problem statement

Sri Lanka has been a developing country for decades, even after gaining independence in 1948. The country's economy was further impeded and impotent to achieve sustainable growth by the war between the Sri Lankan government and the Liberation Tigers of Tamil Eelam (LTTE) for nearly three decades, one of the longest running civil wars in Asia. The latter can be considered as one of the major incidents that had a massive adverse impact on Sri Lanka's economic growth caused by the rise of inflation due to the collapse of the tourism industry (as due to the war, Sri Lanka was perceived as an unsafe destination), the decline in productivity due to the inefficiency of the manufacturing industry in Sri Lanka as well as the high cost of defense.

Since the end of the civil war in Sri Lanka after the period of 2008 and 2009, the highest inflation rate 12.1% was reported in December 2021. The reasons behind the rising inflation are the depreciation of the Sri Lankan Rupee (LKR), cost-push inflation due to rising oil prices, excessive printing of money by the Central Bank of Sri Lanka (CBSL) that caused rise in inflation (as the money supply plays a more important role in determining prices), increase in the price of goods and services in the economy, and the eroding purchasing power of money. Moreover, agriculture production capacity declined since farmers halted cultivation, partly due to not achieving the harvest as expected (due to the government ban of importing chemical fertiliser to encourage farmers to use local organic fertilizers causing resistance among farmers) and imposing sanctions on imports in the face of the global COVID-19 pandemic. In addition, this situation worsened due to levying one-third of imported goods by the government as well as the devaluation of LKR against the appreciation of United States Dollar (USD) which in turn lead to higher prices on imported goods (such goods becoming expensive), thus contributed to inflation growth.

According to the CBSL, in 2000 and 2021, Sri Lanka's inflation rate remained at 6.18% and 5.92% respectively. This indicates that Sri Lanka's inflation has been steadily fluctuating but with no significant decrease during this period. Furthermore, Sri Lanka's economic growth rate of 6% in the year 2000 declined to -3.57% in the year 2021. Accordingly, it appears that Sri Lanka's economic growth has fluctuated significantly reaching negative growth during the end of the above mentioned period. Thus, this study investigates the impact of inflation on the slowdown in economic growth in Sri Lanka.

## Objective

This study aims to investigate the impact of inflation on economic growth in Sri Lanka. As a consequence, this study differs from the existing empirical studies and it contributes to the literature in three ways. Firstly, the present study used quarterly data for a long period of time and employed Auto Regressive Distributed Lag (ARDL) model to conduct. Using quarterly data rather than annual data provides the ability to identify and monitor the impact of inflation

on economic growth from frequent points. In addition, enriching the existing literature, the present study will further be an immense benefit for any researcher or stakeholder in this subject area to gain more insights.

Secondly, considering the past literature, most studies have examined the relationship between inflation and economic growth. A few studies have investigated the impact of inflation on economic growth, whereas studies of this kind have not been recently conducted based on data considering two decades during 2000–2021 contributes to the empirical gap. Moreover, this study is timely due to the behaviour of inflation and economic growth in the country at present, specially due to the economic instability triggered by the COVID-19 pandemic.

Finally, the present study contributes to modifying policies to improve the existing economic situation while providing recommendations to the government and other economic decision makers and allows ascertaining the effectiveness of decisions and policies by policymakers so far on the behaviour of inflation and economic growth in Sri Lanka. Consequently, this study is significant and worthwhile to conduct.

Furthermore, the rest of the study consists of five main sections, each providng a detailed overview of this study. The second section discusses the literature review and underlying the significance of this study, the third section discusses the data and methodology and the empirical findings are evaluated in the fourth section. The fifth section consists of concluding remarks with policy implications and recommendations.

## Literature review

In the preliminary stage of this study, researchers investigated research papers from various reliable sources while ensuring quality. As depicted in Fig 1, research papers were selected by referring to Science Direct, Wiley Online, Emerald Insight, JSTOR, Springer, Sage Publications, Research Gate, and Google Scholar and the search terms used were: impact or relationship, inflation and economic growth, Gross Domestic Product (GDP), and ARDL. Further, 197 publications were initially identified through the above sources and excluded 32 publications in the screening process due to unsatisfactory/irrelevant or insufficient information and lack of relevance.

The remaining 165 research papers were subjected to a second screening procedure; 136 publications were selected based on the title and relevancy of the abstract. At this stage of screening, 116 publications were identified based on key terms and text to find the most relevant publications for this study. Thereafter, this study was limited to 80 publications as the remainder were not contextually relevant. As a result, it was realised through the process that the remaining 56 research papers were eligible for conducting; moreover, these can be referred to as quality publications to accomplish the purpose of the study regarding the impact of inflation on economic growth in Sri Lanka.

Inflation is a key macroeconomic factor in each and every economic reality. According to McConnell, Brue [5], the rise in the overall price level is referred to as inflation and in a situation where inflation is high, the purchasing power of goods and services is declining. Previous research has shown that inflation has a significant impact on the economy [6]. Moreover, the stability of a country depends on the behavior of inflation and by monitoring those fluctuations, government get the opportunity to maintain the inflation through fiscal and monetary policy. Cost-push inflation and demand-pull inflation are the two types of inflation identified by Mankiw [7]. Demand-pull inflation is generated by the increase in aggregate demand where cost-push inflation resulting because of increase of the price and decrease in output levels. According to McConnell, Brue [5], Sri Lanka is now experiencing cost-push inflation,

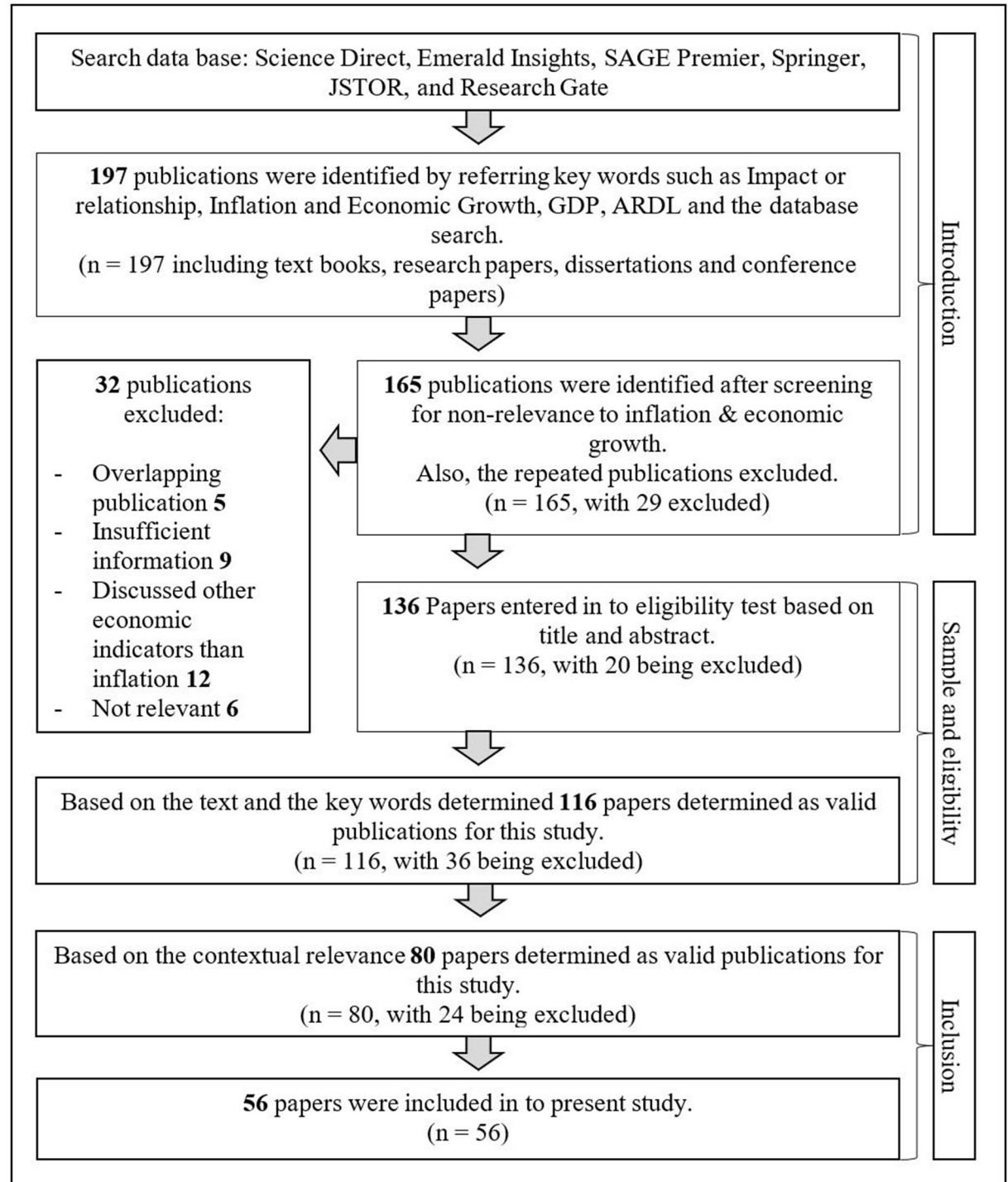

**Fig 1. Literature search flow diagram.** Source: Based on authors' observations.

which the CBSL interprets as temporary due to supply chain disruptions and rising global commodity prices.

Scholars have deliberated inflationary impacts from various perspectives. According to existing literature, these have been tested in developed and developing countries and factors

that influence on inflation. Many studies discovered that the inflation as one of the macroeconomic factors that affect on economic stability [8]. The effect of persistent high inflation rate both in the long run and short run was investigated in similar studies by Faria and McAdam [9]. Lopez-Villavicencio and Mignon [10] and Kankpeyeng, Maham [11] examined the impact of inflation on economic growth in diverse groups of countries, including both developed and developing economies. In addition, Gregorio [12] observed a long-term relationship between inflation and economic growth using data from 12 Latin American countries over the 1951–1985 period.

Empirical studies unveiled some exciting and essential findings of inflation impacting economic growth. Furthermore, it is vital to be vigilant on concerns about the changes in inflation and economic growth in other countries. As Risso and Carrera [13] and Uddin [14] pursued, the 9% of inflation rate is higher than the expected level of the economic progress of Mexico. In such situations, one must analyse the reactions of inflation and adopt suitable policies to avoid such unintended consequences that cause massive impacts. Moreover, Marinakis [15] has carried out a study by using statistics from Latin America during 1988–1991 and proved that all successive attempts of stabilisation failed in Latin America because of the inflation which faced a crisis in 1980s. However, Van [16] has used economic theories such as econometric model, Fisher and Freidman theories to analyse inflation and its effects on the economic growth of Vietnam. In addition, Baharumshah and Soon [17] conducted a study in relation to the Malaysian economy and revealed that the upward inflation decreased uncertainty but on the contrary, economic instability decelerated the output growth in the economy.

Since inflation and economic growth deal with time series data, the analysis can be conducted based on figures on the financial year. Hwang and Wu [18] and Khieu [19] found that when inflation rises by 1%, economic growth slows by 0.61% and when inflation declines by 1%, the economy grows by 0.53%. This further means that a high inflation scenario can be more destructive than the benefits gained from a low inflation scenario. Economists agreed that significantly lower but positive inflation contributes to economic growth, which in turn enables macroeconomic policy to maintain price stability and achieve sustainable economic growth. In general, inflation was deliberated as a crucial issue in sustainable economic growth, because the latter is a key factor in affirmative economic health as pointed out by Girdzijauskas, Streimikiene [20].

For analysis in Taiwan and Japan, Lee and Wong [21] referred the regression model to investigate the link between the inflation threshold and economic growth. As presented by Hoang [22] and Ball [23], in the long run, an increase in the inflation rate has a greater impact than a decrease and severe inflation will inflict damage on economic activity. The empirical findings denoted that economic growth will enhance when inflation is lower in nature. In contrast, Faria and Carneiro [24] and Akinsola and Odhiamb [25] confirmed that inflation does not affect the real output in the long run and it does have a negative impact on real output in the short run. Further, Akinsola and Odhiamb [25] indicates that inflation affects economic growth over time due to changes in the charasteristics of developed and developing countries. In addition to these studies, the relationship between inflation and economic growth has been hypothesized based on unilateral factors. As stated by Sapic, Obradovic [26], there is an unblemished linkage between inflation and economic growth in Serbian economy.

Besides, a few similar studies were conducted based on the context in Sri Lanka. The negative relationship in long run and substantial correlation between inflation and economic growth in Sri Lanka was scrutinised by Madurapperuma [27] and Bandara [28].

Other than inflation, some other factors significantly affect economic growth. Khan [29] stated that apart from inflation, many factors affect economic growth, such as the monetary policy as a key economic factor and unemployment rate. Armantier, Kosar [30] has conducted

a research to investigate how the New York Federal Reserve's consumer expectations survey reported inflationary beliefs during the first six months of the COVID–19 outbreak. During this pandemic, consumer expectations and inflation uncertainty increased. The rise in inflation uncertainty has been attributed to the high rate of precautionary savings.

Feng [31] conducted a study to examine the relationship between China's political events and macroeconomic performance. By reviewing the results, it can be stated that under previous political leaderships, the economic progress has been stable and the unemployment and inflation was low during the years the communist party has been in the power. According to this study, regardless of many variables that affect economic growth in Sri Lanka, the main concern should be inflation.

However, the analysis can be conduct based on many analytical tools and methodologies to find out the best outcome. In their study, Burdekin, Goodwin [32] referred a panel estimate of the impact of inflation on economic growth. Chaudhry, Ayyoub [33] employed the annual time series data during 1972–2010, researched using the Ordinary Least Square method and analysed whether inflation encourages the economy. These findings determined that the rise in inflation has been attributed to the appreciation of the currency and the deficit.

Besides, Oikawa and Ueda [34] established an endogenous growth model, which was referred to investigate the link between inflation and economic growth. The model calibration demonstrates that the optimal inflation rate is close to the growth-maximising inflation rate, and that its divergence has significant consequences. In addition, Huybens and Smith [35] developed a monetary growth model indicating that inflation is inversely associated with the financial market, influencing economic growth. According to Aydın, Esen [36], the panel data threshold analysis proved a nonlinear relationship between inflation and growth rate. Further, Khoza, Thebe [37] used the Smooth Transition Regression model to establish the monotonic link between inflation and economic growth in South Africa. Thus, the findings confirmed that the 5.3% of inflation threshold caused a negative impact on economic growth over this level.

In order to have a better understanding on the impact, it is vital to identify whether there is a significant relationship between inflation and the economic growth. In this sense, the study referred to many tests including the ARDL test to examine both long run and short run impacts of inflation on economic growthto evaluate the effect of a government debt ceiling on Africa's economic growth. Using the ARDL model, Mandeya and Ho [38] and Nadabo and Maigari [39] explored that inflation has a negative impact on growth both in the short and long run, and the inflation uncertainty in South Africa is a short-term phenomenon with no long-term implications.

Additionally, a series of steps are to be followed in ARDL bounds test. The study based on bound testing approach developed by Pesaran, Shin [40] indicated that no long-run relationship exists in six countries except in one country. Here, researchers concluded that most countries have a short run relationship between inflation and economic growth. Furthermore, Manamperi [41], Datta and Mukhopadhyay [42] explored a significant short-run relationship but not a long-run relationship between inflation and economic growth.

According to Pinshi [43], the Error Correlation Method (ECM) estimates that the dependent variables return to equilibrium after the other variables change. ECM can be used to examine the short- and long-term effects of exchange rate changes on inflationary behaviours based on the p-value. Further, Phiri [44] has employed both thresholds ECM and Granger Causality analysis to ascertain the relationship between the economic growth and financial development. Autocorrelation is the similarity between a time series and a lagged version over a succession period of time. Durbin Watson (DW) has used to evaluate serial correlation errors as a test to detect autocorrelation. It is a complex test, but has demonstrated numerous

regression analysis that are relevant in certain situations. As per Hajria, Khardani [45], the Breusch Godfrey test for the ARDL model evaluates the correlation and thereby concerns the quality of the compatibility. According to Mohseni and Jouzaryan [46] and Sapic, Obradovic [26], Cumulative Sum (CUSUM) test is referred to check the stability of the selected method. The structural changes can be assessed by monitoring statistics and graphical illustrations perceived through CUSUM.

Additionally, policy implications in the country can directly affect the economy when inflation exaggerates economic growth as a macroeconomic factor. According to Chowdhury [47], macroeconomic factors of Indonesia is crucial when declaring policies related to their economy. The findings implied that the policies will sustain social spending and relieve the government of debt, and economic recovery would not be stymied by tightening immature monetory policy. Moreover, numerous studies have revealed that the impact of inflation on economic growth led to significant changes in policies. As per Rhys and Barry [48], an increase of 1% in inflation is likely to cause 77% rise in Malaysia's economic growth rate which caused changes in policies.

The findings of Mendonca [49] has illustrated that the inflationary process implemented in Brazil after following the inflation target is not related to the emergence of output-inflation trade off. As inflation rises, governments adopt a variety of financial restriction tactics to safeguard specific sectors of the economy. Accordingly, establishing maximum interest rates on deposits and loans, restricting credit supplied to certain economic activities, and taxing the earnings of financial intermediaries are some of financial restriction tactics employed [50, 51]. An increase in inflation to a particular level reduces both the amount of savings and the number of savers, according to a theoretical research by Moore [52], Azariadis and Smith [53], Choi, Smith [54]. Thus, an increase in inflation reduces the amount of credit available in an economy.

Most empirical studies were based on developed countries, where relevancy of such findings in the context of developing countries is questionable. This paper is pioneer in examining the impact of inflation on Sri Lanka's economic growth as a developing country.

## Data and methodology

### Data

Objective of this study is to scrutinise the impact of inflation on economic growth in Sri Lanka. Data related to variables were collected and determined through time series data. This research is totally based on quantitative data gathered through the CBSL and the Department of Census and Statistics (DCS) from Q1 2000 to Q4 2021. The impact can be examined by analysing data related to inflation rate (INF) as the independent variable and GDP as the dependent variable on economic growth. STATA analysis tool was used to estimate the model and this study provides Data file for this analysis in S1 Appendix.

### Methodology

Time series variables and its changes can be analysed by using an econometric model. Research approach for the present study is designed to inquire the correspondence between inflation on economic growth which investigates whether there is a positive or negative impact on economic growth through inflation of Sri Lanka; in addition, significance of the long run and short run impact is also ascertained. In this research, lags for the variables are considered by using exclusive sources and has not captured ethical issues.

In addition, data points are indexed based on time and the present research is to be conducted through the time series analysis, that also enhances reliability of the research. As the

econometric model, ARDL model will evaluate the long-term as well as short-term impacts of inflation on economic growth. This model also allows investigating lags between variables. The general equation for the ARDL model can be expressed in (1):

$$\Delta Y_t = \beta_0 + \sum_{i=1}^{n} \beta_i \Delta y_{t-i} + \sum_{i=0}^{n} \delta_i \Delta x_{t-i} + \varphi_1 y_{t-1} + \varphi_2 x_{t-i} + \mu_t \qquad (1)$$

Explanation:
Dependent variable–Y
Independent variable–X
Short-run coefficients–$\beta_i$, $\delta_i$
ARDL long-run coefficient–$\varphi_1$, $\varphi_2$
Disturbance (white noise) term—$\mu_t$

 ARDL model examines the correlation between the variables and the stability of the referred variables must be investigated when conducting the research. According to Shahid [55], the consequences of inflation on growth of the Pakistan economy were examined through time series data and the same application was referred in this research. In future, based on the results generated through the selected methodology, the responsible authorities can endeavour to amend policies that support economic growth and for stability of macroeconomic factors such as inflation.

## Results and discussion

The following findings indicate the extend of the impact of inflation on economic growth in Sri Lanka. Firstly, fluctuations in both inflation and economic growth for the selected time period should be evaluated. These are depicted through statistical tables and graphical illustrations.

 The above graph shows the significant changes in both INF and GDP over the period of Q1 2000 and Q4 2021 in Sri Lanka. Fig 2 illustrates that INF rises rapidly 2.2% to 15.3% from Q1 2000 to Q4 2001, along with the marginal rise in GDP from 339,578 million USD to 368,461 million USD. Moreover, in the Q4 2021, INF rises to 9.9% and the GDP is likely to slow down from 2,497,489 million USD to 2,169,203 million USD. Through this analysis, a better understanding can be gained about how INF has obstructed GDP, and hence the impeding economic growth prospects. Therefore, when considering the impact of INF on GDP, this study can be expanded.

### Stationary test

To examine the impact of INF on GDP in Sri Lanka, the overall regression results might be fabricated and testing for stationarity as the first step is significant. In case the data set is not stationary, it is impossible to generate results which are useful and can be used in usual econometric procedures.

 According to Fig 3, both (a) GDP and (c) INF can be identified as time series in which the situations and statistical properties change over time and it can be concluded that both variables are non-stationary. However, as (b) and (d) time series lines are drawn on the first difference, it can be inferred that GDP and INF are stationary with statistical properties and instances that change over time to some extent. Further verification of the stationarity of these two variables can be investigated on the basis of statistical evidence, by employing Augmented Dickey Fuller (ADF) test and Dickey Fuller for Generalized Least Square (DF-GLS) test.

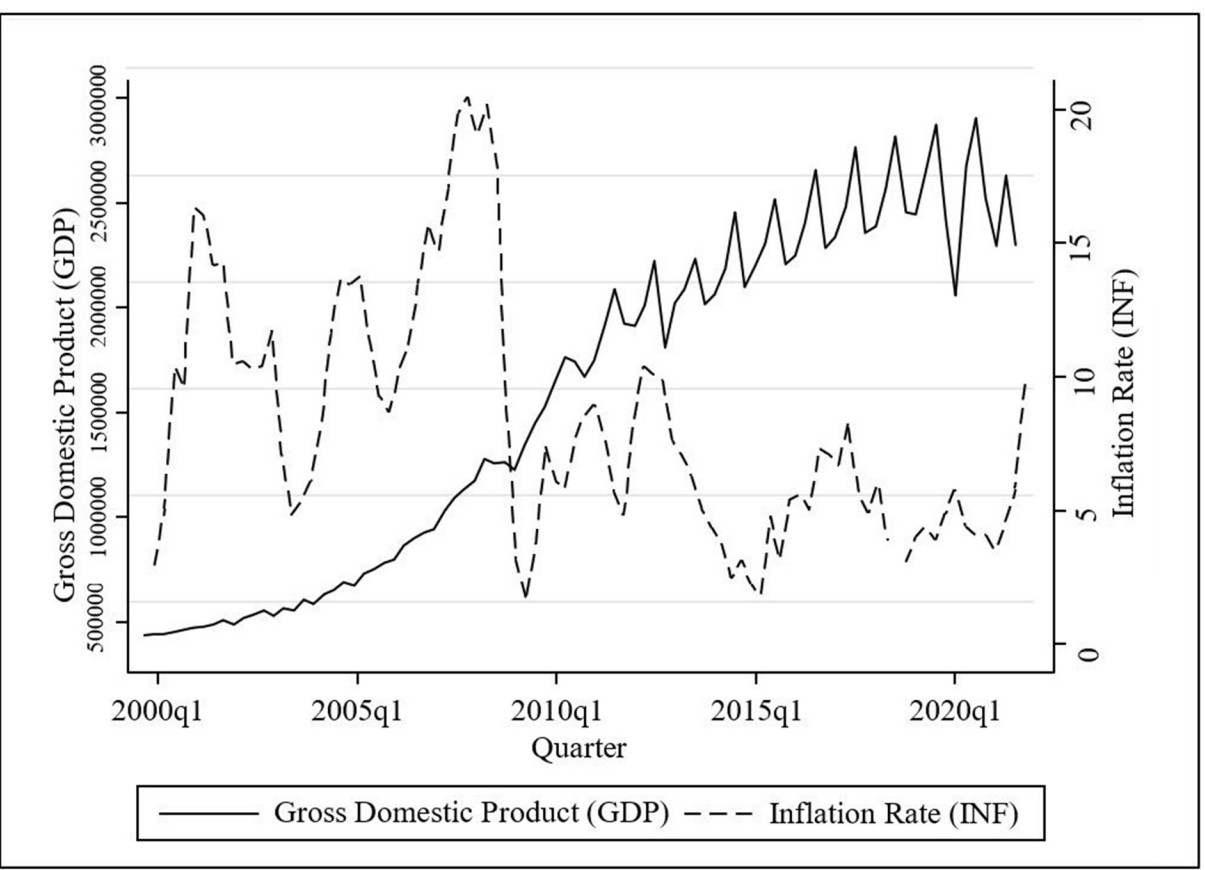

**Fig 2. Graphical representation of changes between inflation and economic growth.** Source: Authors' illustration based on CBSL [56] and DCS [57].

The unit root test is used to assess the time series data's stationarity properties, as the findings provided from regression models have the possibility to generate spurious results [42]. According to the ADF results of GDP and INF in Table 1, the MacKinnon approximate *p*-value is higher than the 5% critical value. Therefore, GDP and INF can be considered as non-stationary variables and the unit root null hypothesis cannot be rejected at that level of significance. According to Hoang [22], the time series that regressive outcomes can be distorted when the estimated variables are non-stationary. Moreover, since the *p*-value of GDP and INF at first differences (ΔGDP and ΔINF) are 0.0000, nonstationary null hypothesis is rejected and concluded that the differences of the GDP and INF are stationary.

Furthermore, results of the ΔGDP and ΔINF of DF- GLS test, the values of tau t-statistic were attained as -11.18 and -4.88 respectively. Since the null hypothesis of unit root can be rejected at all significance levels, the time series is stationary. In accordance with studies conducted by Pesaran, Shin [40], the ARDL model can be employed since the variables are I(0) or I(1). Based on the above results, all the variables are integrated and stationary at I(1) sequence, which allow the study to examine the impact of inflation on economic growth in Sri Lanka.

## ARDL bound test for co-integration

After determining the order of integration of the macroeconomic variables used in this study, ARDL bounds test is to be assessed. This is to evaluate the long-run co-integrating relationship among the variables and thereby, to identify whether the ECM should be run or not.

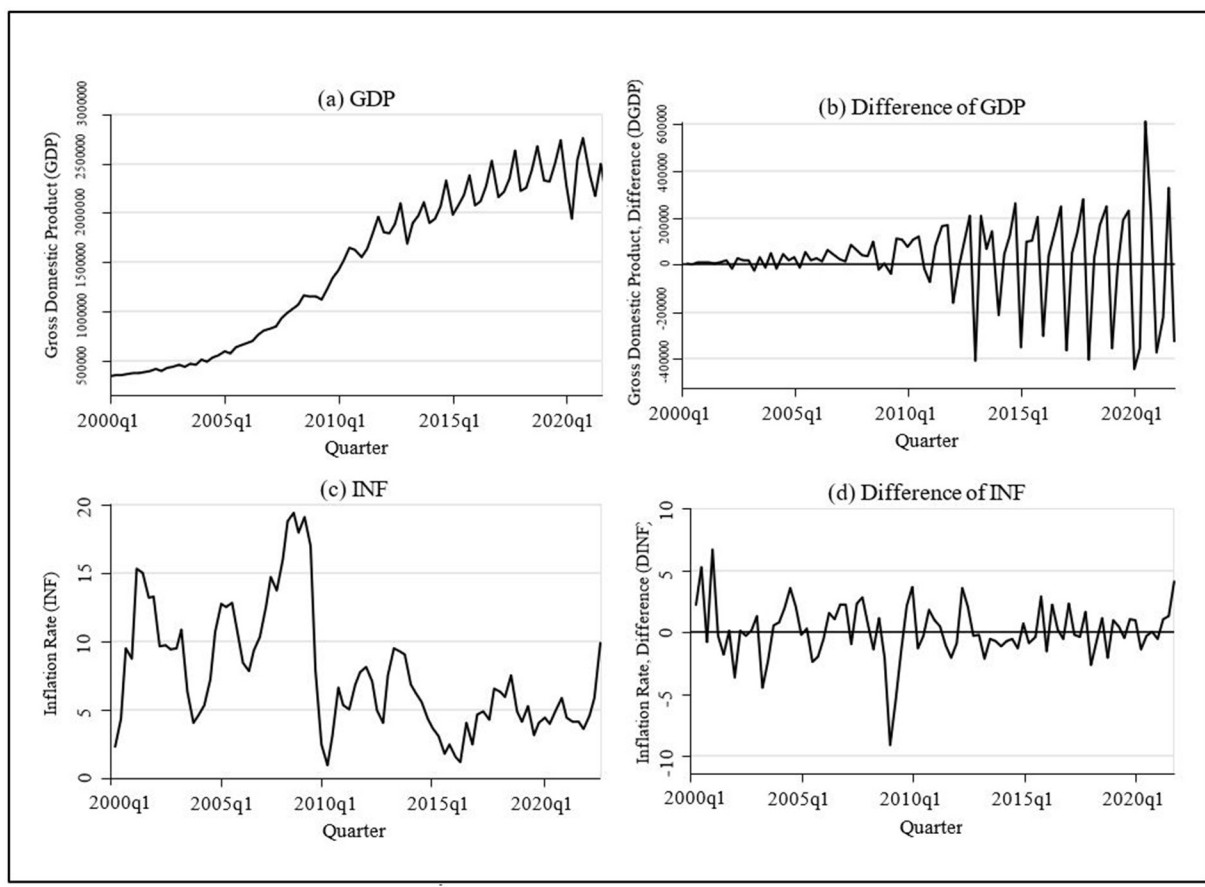

**Fig 3. Time series plot to check stationary.** Source: Authors' illustration based on CBSL [56] and DCS [57].

After determining the optimal lag length the Akaike Information Criterion was employed and it indicated that the maximum number of optimal lags for ARDL is 5,0 for GDP and INF, respectively. This implies that the ARDL model fits well into this dataset, and this provides the ability to identify the number of periods that impact economic growth and also enhances quality of the study.

As Pesaran, Shin [40] has presented, ARDL bound test for co-integration provides evidence for cointegration since the F statistic is greater than the critical values. As shown in Table 2,

**Table 1. Unit root tests for stationarity–GDP and INF.**

| | Augmented Dickey Fuller—Unit Root | | Dickey Fuller—Generalized Least Square | |
|---|---|---|---|---|
| | *p*-value | Critical value | T Statistics | Critical Value |
| GDP | 0.6863 | -2.9012** | -3.0681 | -3.0663** |
| INF | 0.0207 | -2.9012** | -2.9103 | -2.7722* |
| ΔGDP | 0.0000 | -1.9504** | -11.1778 | -3.6291*** |
| ΔINF | 0.0000 | -1.9504** | -4.8802 | -3.0686** |

Notes:

*, ** and *** indicate significance at 10%, 5% and 1% respectively; Δ is the first difference.

Source: Authors' compilation based on CBSL [56] and DCS [57].

**Table 2. Results of ARDL bound test and ECM.**

| ARDL Bound Test | | |
| --- | --- | --- |
| **Boundaries** | **F Statistic (1.6062)** | **t Statistic (-1.7921)** |
| I(0) | 4.9412** | -2.8625** |
| I(1) | 5.7341** | -3.2216** |
| **ECM Test** (R-squared: 0.6793) | | |
| **Regressors** | **Long Run** | **Short Run** |
| GDP | -107263.8256 | -3427.9352 |
| INF | 0.5552 | 0.6362 |

Notes:

** indicates significance at 5%, I(0) indicates lower bound and I(1) indicates upper bound.

Source: Authors' compilation based on CBSL [56] and DCS [57].

the value of F statistic is 1.6 which is higher than the critical values of 5%, concludes there is a co-integration between GDP and INF. Since there is co-integration, the study proceeds with model estimations using the ARDL bound test and ECM approaches. As stated by Mandeya and Ho [38] and Manamperi [41], before analysing impact of INF on GDP, it is essential to examine the relationship between these two variables. According to ARDL bound test results, the t statistic is (-1.8) and the critical value of upper bound is less than that (-3.2) at significant level of 5% and the null hypothesis can be rejected. Furthermore, this indicates that there is a short run relationship between GDP and INF; it shows that when INF increases, GDP declines, but it does not remain the same for a long time. Similarly, it can be further explained that the relationship between GDP and INF diminishes over time and hence the relationship exists in the short run.

ECM directly assesses the sensitivity of the dependent variable after the other independent variables have changed [43]. According to ECM results provided in Table 2, there is a negative significant coefficient and a long-term impact in the time series; the findings illustrate that 67% variation in the GDP have been explained by the INF. It further revealed that whenever the INF increases by 1%, the GDP of Sri Lanka falls by 107,263.82 million USD in the long run. Theoretically and empirically these findings are further linked to those discovered by Ball [23]. As such, these reconfirm that INF will have a significant impact on GDP in Sri Lanka in long run.

In the short run, when INF increases by 1%, GDP decline by 3427.94 million USD. This suggests that there is a negative impact of INF on GDP in Sri Lanka in the short run, and Faria and Carneiro [24] further justify this phenomenon. Thus, INF remains as a significant negative factor forGDP, and INF has become a critical variable in short and long term impeding GDP. This suggests that GDP has the potential to adjust to a long-term equilibrium after the short-term shocks created by INF. Further, it highlights that GDP is particularly affected by extraordinary extensive INF in Sri Lanka.

## Normality and stability tests for autocorrelation

Based on the distributed sample size, this research directed to choose DW test to check after a regression analysis which test autocorrelation in the residuals from the statistical model.

According to the Table 3, 1.6513 of DW d-statistic test refers, INF is having positive influence on GDP. As per Hendry, Mizon [58], there should not be any exogenous cause to independent variable INF. If there is a positive serial correlation of INF in the previous quarter, it is likely to cause positive correlation on GDP in the next quarter, which will vital to determine

**Table 3. Results of post estimation diagnostic and stability tests.**

| Durbin-Watson Statistic Test (Ho: No serial correlation) | Breusch-Godfrey Test (Ho: No serial correlation) | CUSUM | CUSUMSquared |
|---|---|---|---|
| 1.6513** | 0.4348** | S | S |

Notes:

**indicates significance at 5%; S indicates stability.

Source: Authors' compilation based on CBSL [56] and DCS [57].

changes in GDP much proactively and be geared for necessary amendments in policies to avoid unrealistic changes.

Secondly, Breusch-Godfrey LM test will estimate serial correlation based on the number of lags unlike in basic autocorrelation tests. Lags will determine through autocorrelation between residuals of ARDL. Since the *p* value of LM test is 0.4348 and it is less than 5%,the null hypothesis can be rejected and there is no serial correlation between residuals. However, Hajria, Khardani [45] stated the goodness-of-fit examining Breusch-Godfrey LM test in Autoregressive models. If the null hypothesis is rejected, there is autocorrelation in the residuals. Negative serial correlation can be noted and such variables should not be over differenced for the purpose of maintaining a quality analysis.

## Cumulative sum for autocorrelation

The stability of the model can investigated through CUSUM test based on the residuals of the selected data points. The former tests were conducted systematic changes of INF and GDP. Many researchers took advantage of ARDL model to achieve their research objectives, and the CUSUM test proved the model's stability in advance [46].

The stability of short and long run coefficients is analysed through CUSUM test [26]. In addition, CUSUM test ensures whether acceptable outcomes are being achieved and can be determined through stability of the ARDL model. Fig 4 demonstrates that the calculated model meets the stability criteria with no roots outside of the significant limit. Accordingly, the coefficient of the long run ARDL is consistent with the CUSUM squared test findings. The model parameters indicates that the model is stable, whereas the statistics lying within the boundary line means the outcomes of the ARDL test are accurate.

Overall, the ADF for unit root test demonstrated that series for ΔINF and ΔGDP is stationary. Further, ARDL bound test was referred to investigate the existence of the long run and short run relationship to determine the impact by employing ECM test. The short run relationship was concluded through ARDL bounds test and the significant impact in long and short run was determined in ECM test. Finally, the stability of the ARDL model assessed through CUSUM test where it reclined at 5% boundary. Therefore, all the test results revealed that there is an impact of Inflation on economic growth in Sri Lanka.

## Conclusion

This research study explores how inflation impacts on economic growth in Sri Lanka over the time period from Q1 2000 to Q4 2021. By utilising the ARDL model, the full sample of this study inspected that there is a short run relationship between inflation and economic growth, respectively; further, analysis reveals that while inflation increases, GDP declines but this impact does not remain the same for a long time. Based on the findings of the ECM test, it can be concluded that inflation disrupts economic growth and there is a significant negative impact of inflation on economic growth in Sri Lanka in both short run and long run. This

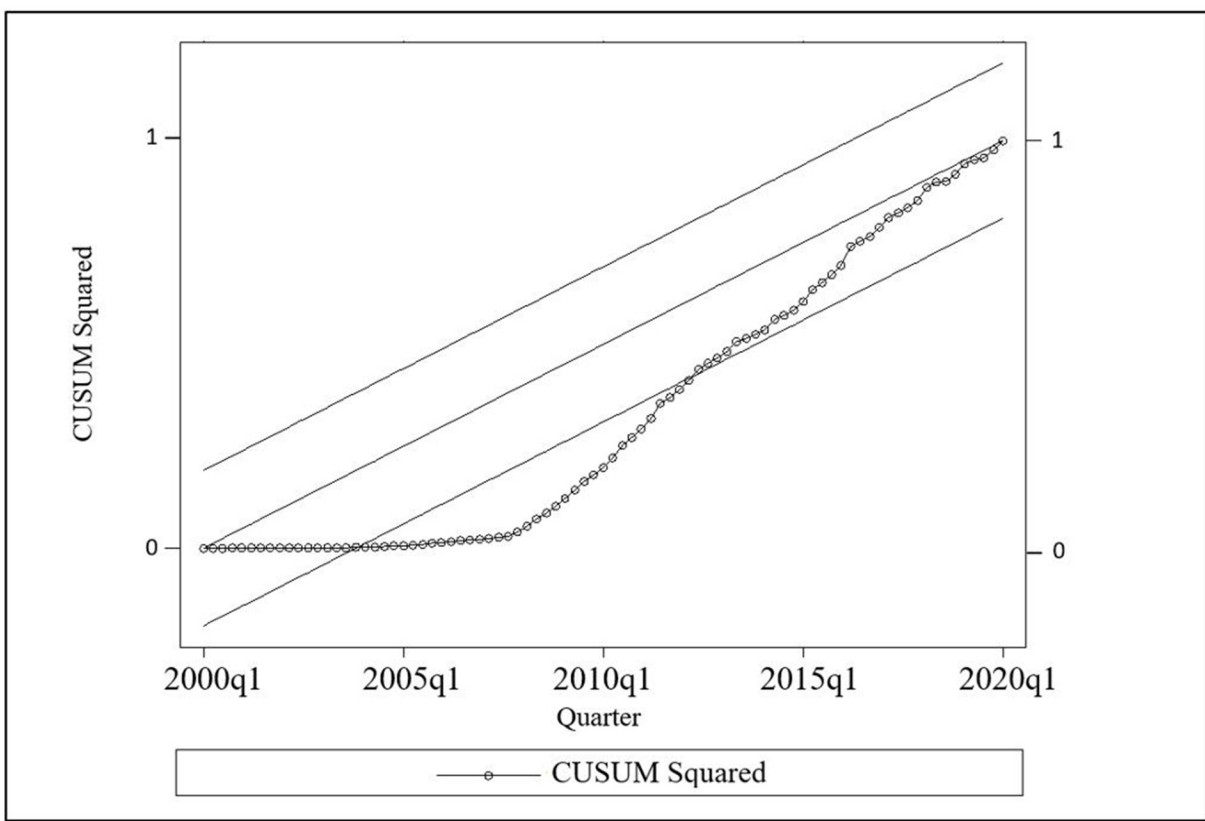

**Fig 4. Results of CUSUM squared.** Source: Authors' illustration based on CBSL [56] and DCS [57].

implies that inflation has become a critical factor in influencing economic growth. As demonstrated by the CUSUM test results, the residuals were typical, constant, and mostly stable. The findings implies that the economy should place more emphasis on the significance of inflation on economic growth and ensure price stability to create a conducive economic environment for short and long-term growth.

Further, available literature indicates mixed results and somewhat contrary in developing countries, with inflation having adverse impacts than in developed countries. In future, the scope of studies is worth expanding to cover the impact of economic shocks, recessions and booms and extraordinary events of high magnitude such as pandemics of this nature in both developed and developing countries. These findings can provide a big picture of the many potential scenarios that can occur due to many fluctuations in the economy—locally, regionally and globally and thus calls for adaptability in pursuit of economic resilience.

## Policy implication and recommendations

This research study demonstrates that the rise of the price level or the inflation, has a negative impact on both in the long run and short run on the economic growth in Sri Lanka. Consequently, these findings have significant policy implications (fiscal and monetary policies) for both local policymakers and macro-economists, emphasising that the stability of the inflation rate is a prerequisite for enhancing the economic growth in Sri Lanka. Generally, this study seeks for a much clear explanation regarding the increase or decrease in the average price level has a significant impact on the country's economic growth. Findings can be critical in

investigating the impact of essential goods in Sri Lanka, which are mostly imports. Accordingly, it can be ensured that maintaining a low inflation rate in the long run is conducive to boosting the country's economic growth, living standards and for a sustainable economy. Furthermore, this study identifies the depreciation of the LKR against the USD rising oil prices, increased tendency to print local money, higher government expenditure, lower agricultural production, and increases in the prices of essential goods and services as the main reasons for the rapid rise in inflation in Sri Lanka. This is due to the market's demand for products and services grows rapidly relative to the existing supply in the country. Furthermore, the contentious policy decision of the Government of Sri Lanka to ban the use of chemical fertilisers for cultivation in 2021 has led to agriculture supply disruptions resulting in a surge in the prices of goods and services in the economy. Accordingly, as a response to this economic concern, local policymakers should focus more on the development of agricultural sector in the country through the development of agricultural infrastructure. Another essential concern is to develop the manufacturing sector, where most import substitutes can be produced locally, i.e. supply deficiencies will be much less, hence causing less inflationary pressures. As such, this study recommends that the policymakers should act proactively and in future take efforts to control the country's current high level of inflation and implement policies that stimulate constant economic growth.

## Supporting information

**S1 Appendix.**
(XLSX)

## Acknowledgments

The authors would like to thank Ms. Gayendri Karunarathne for proof-reading and editing this manuscript.

## Author Contributions

**Conceptualization:** Piumi Atigala, Tharaka Maduwanthi, Vishmi Gunathilake, Ruwan Jayathilaka.

**Data curation:** Piumi Atigala, Tharaka Maduwanthi, Sanduni Sathsarani.

**Formal analysis:** Piumi Atigala, Tharaka Maduwanthi.

**Methodology:** Vishmi Gunathilake, Ruwan Jayathilaka.

**Resources:** Piumi Atigala, Tharaka Maduwanthi, Sanduni Sathsarani.

**Software:** Piumi Atigala, Tharaka Maduwanthi, Vishmi Gunathilake, Sanduni Sathsarani.

**Supervision:** Ruwan Jayathilaka.

**Validation:** Piumi Atigala, Vishmi Gunathilake, Ruwan Jayathilaka.

**Visualization:** Piumi Atigala, Tharaka Maduwanthi.

**Writing – original draft:** Piumi Atigala, Tharaka Maduwanthi, Vishmi Gunathilake, Sanduni Sathsarani, Ruwan Jayathilaka.

**Writing – review & editing:** Ruwan Jayathilaka.

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
