## [Decision Letter · Decision Letter 0]

18 Jul 2022

PONE-D-22-06608Driving the Pulse of the Economy or the Dilution Effect: Inflation Impacting Economic GrowthPLOS ONE

Dear Dr. Ruwan Jayathilaka,

Thank you for submitting your manuscript to PLOS ONE. After careful consideration, we feel that it has merit but does not fully meet PLOS ONE’s publication criteria as it currently stands. Therefore, we invite you to submit a revised version of the manuscript that addresses the points raised during the review process.

We look forward to receiving your revised manuscript.

Kind regards,

Ricky Chee Jiun Chia

Academic Editor

PLOS ONE

Journal Requirements:

Reviewers' comments:

Reviewer's Responses to Questions

**Comments to the Author**

1. Is the manuscript technically sound, and do the data support the conclusions?

Reviewer #1: Yes

2. Has the statistical analysis been performed appropriately and rigorously? 

Reviewer #1: Yes

3. Have the authors made all data underlying the findings in their manuscript fully available?

Reviewer #1: Yes

4. Is the manuscript presented in an intelligible fashion and written in standard English?

Reviewer #1: Yes

5. Review Comments to the Author

Reviewer #1: The length of the paper could be reduced and written in a more concise manner.

6. PLOS authors have the option to publish the peer review history of their article (what does this mean?). If published, this will include your full peer review and any attached files.

Reviewer #1: No

---

## [Author Response · Author response to Decision Letter 0]

30 Jul 2022

Point–by–point response to reviewers

Dear editor and reviewer.

Greetings. Thank you very much for the valuable and effective comments.

Comments of Reviewer 1:

General Comments: The length of the paper could be reduced and written in a more concise manner.

Comments of Authors: Thank you very much for the comment and well noted. Suggestions have been incorporated in the revised version. We are sharing the revised version, which was proofread, and number of words reduced to 7,383 from 8,174.

---

## [Editor Report · Decision Letter 1]

8 Aug 2022

Driving the Pulse of the Economy or the Dilution Effect: Inflation Impacting Economic Growth

PONE-D-22-06608R1

Dear Dr. Ruwan Jayathilaka,

We’re pleased to inform you that your manuscript has been judged scientifically suitable for publication and will be formally accepted for publication once it meets all outstanding technical requirements.

Kind regards,

Ricky Chee Jiun Chia

Academic Editor

PLOS ONE
---

## [Editor Report · Acceptance letter]

10 Aug 2022

PONE-D-22-06608R1 

Driving the Pulse of the Economy or the Dilution Effect: Inflation Impacting Economic Growth 

Dear Dr. Jayathilaka:

I'm pleased to inform you that your manuscript has been deemed suitable for publication in PLOS ONE. Congratulations! Your manuscript is now with our production department. 

Kind regards, 

on behalf of

Dr. Ricky Chee Jiun Chia 

Academic Editor

PLOS ONE